# Facilitating the access to HIV testing at lower costs: "To the laboratory without prescription" (ALSO), a pilot intervention to expand HIV testing through medical laboratories in France

Karen Champenois[1]*, Victoire Sawras[1], Pamela Ngoh[2], Philippe Bouvet de la Maisonneuve[3], Julie Valbousquet[3], Margot Annequin[2,3], Yoana Gatseva[1], David Michels[4,5], Nathalie Lydié[6], Charlotte Maguet[2], Elodie Aïna[2], Erwann Le Hô[7], Eve Plenel[2], Irit Touitou[3], Sylvie Deuffic-Burban[1], France Lert[2], Pascal Pugliese[3], the ALSO group[¶]

1 Université Paris Cité and Université Sorbonne Paris Nord, Inserm, IAME, Paris, France, 2 Vers Paris Sans Sida, Paris, France, 3 CHU de Nice, Corevih, Côte d'Azur University, Nice, France, 4 AIDES, Pantin, France, 5 Laboratoire de Recherche Communautaire, Coalition PLUS, Pantin, France, 6 Santé Publique France, Saint-Maurice, France, 7 Objectif Sida Zéro, Nice, France

☯ These authors contributed equally to this work.
¶ The complete membership of the author group can be found in the Acknowledgments.
* karen.champenois@inserm.fr

**Data Availability Statement:** Data collected cannot be shared publicly because of their sensitive nature and permission for public use of the data was not

## Abstract

### Background

A pilot HIV testing programme, Au Labo sans Ordo (ALSO; "to the laboratory without prescription") was implemented in two French Fast-Track Cities Initiative areas from 07/2019 to 12/2020. ALSO aimed to remove barriers to HIV testing by providing free testing with widespread access through all laboratories, extended opening hours, and no prescription requirements.

### Objectives

Assessing the ALSO programme in terms of testing activity, user characteristics, and costs, compared to other HIV testing offers.

### Methods

Laboratories and STI clinics reported the monthly numbers of tests performed and positive tests. Two short surveys were carried out 12 months apart in people who sought HIV testing. In each offer, the mean costs of HIV testing have been estimated according to negative or positive results using a microcosting approach.

### Results

During the study period, 214/264 laboratories reported performing 38,941 ALSO tests that accounted for 7.2% of laboratory HIV testing activity. Positivity rates of ALSO and prescribed tests were similar (2.2/1000) but lower than that in STI clinics (6.0/1000).

obtained from the participants. Data are stored on a secure server suitable for hosting health data. Activity and user data are available upon request to the corresponding author or to the data protection officer of the Nice hospital: dpo@chu-nice.fr for researchers who meet the criteria for access to confidential data, after receiving an approval from the ALSO study group.

**Funding:** The ALSO study was supported by Santé publique France (no grant number, https://www.santepubliquefrance.fr) and ANRS MIE (ECTZ118440, https://www.anrs.fr/). The organisations Vers Paris sans Sida, Objectif sida zero and Inserm received the funds. No author has directly received the funds. The funders had no role in study design, data collection and analysis, decision to publish, or preparation of the manuscript.

**Competing interests:** The authors have declared that no competing interests exist.

Heterosexual men, and individuals with multiple sexual partners, poor health insurance and few visits to GPs were more likely to use the ALSO offer than tests upon prescription. Compared to ALSO, STI clinic users were younger, more exposed to HIV and with a less favourable socio-economic situation. ALSO had low costs: €13 for a negative test, €163 for a positive test and €5,388 to identify an HIV-positive person (versus €9,068 in STI clinics and €20,126 with prescribed tests).

## Conclusion

ALSO has attracted users less likely to visit STI clinics or to seek a prescribed test, particularly heterosexual men. Activities, user profiles and costs suggested the complementarity of the HIV testing offers and the relevance of making them coexist. French health authorities have decided to maintain and expand this programme to complement existing HIV testing offers.

## Introduction

Despite innovations and updated recommendations supporting a generalized HIV testing proposal [1], in many countries, testing remains a weak step in the HIV care cascade and might impair the achievement of the end of HIV transmission in 2030. In Europe, 51% of newly diagnosed HIV patients had CD4 counts under 350 cells/mm$^3$ [2].

In addition to promote repeated tests for most exposed people, French HIV testing guidelines recommend to offer HIV tests to any untested individual regardless of exposure issues [3, 4]. General practitioners are the core group of this recommendation as test prescribers. Prescribed tests are conducted in private medical laboratories (hereafter, laboratories) and fully reimbursed by the National Health Insurance. People can get free and anonymous HIV tests in STI (sexually transmitted infection) clinics that mostly operate with walk-in access. Community-based HIV testing, launched in France in 2011, is conducted either by health professionals or trained volunteers with favourable results regarding both acceptability and capacity to reach key populations [5–8]. Self-tests have been approved for sale at community pharmacies since 2015 and for free distribution through outreach programmes since 2018.

An increase in the number of HIV tests was observed in France (+16% from 2010 to reach 6.2 million in 2019), of which more than 70% were performed in private medical laboratories upon medical prescription, 23% in hospitals and 6% in STI clinics [9]. An additional 70,000 tests were carried out by community-based organisations and 79,000 HIV self-tests were purchased in pharmacies. Despite the increased number of tests, the annual number of HIV diagnoses remained stable at approximately 6,000, the number of people living with undiagnosed HIV was estimated to be 24,000, and the estimated time to diagnosis was long (median 3.3 years) [9–11].

Thus, despite a diversity of HIV testing tools and facilities, full reimbursement of HIV tests upon medical prescription and repeated communication campaigns, the HIV undiagnosed population remains too large as regards the first step of the HIV care cascade [12]. Multiple barriers exist on the demand side, among physicians to order an HIV test, and with regard to convenient accessibility. Two systematic reviews assessed barriers to offering HIV testing through healthcare providers in Europe. They highlighted difficulties in addressing HIV issues due to a lack of training to offer HIV tests, disclose the results, and communicate about sexual

health as well as a lack of knowledge regarding HIV testing guidelines [13, 14]. Studies have suggested that removing structural barriers could improve HIV testing uptake by providing convenient, easy-to-access, and free-of-charge testing [13–15].

Paris and Alpes-Maritimes are among the French departments with the highest annual rates of HIV diagnoses and prevalence of undiagnosed cases [9]. Both departments participate in the Fast-Track City Initiative (FTCI) and are committed to reaching the end of HIV transmission by 2030 [16, 17]. Both departments decided to focus on substantially and rapidly expanding the HIV testing volume using existing systems and facilities, i.e. laboratories. Laboratories are well distributed throughout the two areas, are easily accessible with extended opening hours: five full weekdays and Saturday morning. Moreover, STI clinics and community-outreach testing teams were unable to increase their activity due to limitations in dedicated public or charitable funding and a shortage of staff. The French network of laboratories coupled with comprehensive health insurance coverage offers a middle way which may appeal to a range of populations more likely to use a directly accessible service with limited psychological barriers, including pre-test counselling.

The Au Labo sans Ordo (ALSO; "to the laboratory without prescription") programme was launched in partnership with local political and health authorities, professional biologist organisations and the National Health Insurance. The objective of this pilot study was to evaluate, in the two departments where the ALSO programme was implemented, the numbers of HIV tests performed, the characteristics of the populations reached, and the costs of the ALSO programme and compared them to those of other HIV testing offers.

## Methods

### Intervention

The ALSO programme consists of HIV testing performed directly in laboratories, at the user request, without the requirement of a prescription or an appointment and free of charge. The pilot study was conducted in real-life conditions in all the laboratories of Paris and Alpes-Maritimes. Laboratories were reimbursed directly from the National Health Insurance; a specific code was created to allow laboratory payment through usual administrative routines. The ALSO programme was open to all potential users in France, aged 18 or over.

No pre-test counselling was offered, but in agreement with French and European guidelines [3, 18], a minimum of pre-test information was orally provided to the user (voluntary nature of test, confidentiality, details of result delivery). According to national testing guidelines, laboratories performed enzyme-linked immunosorbent assays (ELISAs) and P24 antigen (AgP24) identification in blood samples drawn from the bend of the elbow. Negative results were disclosed to the user, mostly electronically, according to standard procedures to ensure confidentiality. Results were generally available the same day. To compensate for the lack of post-test counselling, negative test results included written information regarding the need to repeat testing after unprotected sex and on pre-exposure prophylaxis (PrEP) as an effective method of preventing HIV. If the test was positive, the laboratory staff called the individual for confirmation testing. Thereafter, people with a confirmed HIV-positive status were referred on a voluntary basis either to their GP or to a dedicated navigation platform able to propose a quick appointment at the HIV clinic of their choice.

Throughout the duration of the pilot, an advertising campaign was conducted in public spaces, on laboratory doors, and in banners on social networks with photos that represented key populations according to gender, age, and origin. The posters contained the following information: "No cost, no prescription, no appointment. Getting tested has never been so easy. HIV tests are offered at all laboratories in Paris/Alpes-Maritimes".

### HIV testing activity

**Activity data collection.** Over the study period, the numbers of HIV tests and positive tests were reported online by each laboratory (monthly) and by local STI clinics.

**Activity statistical analyses.** Descriptive methods were used to assess the use of HIV testing offers, and positivity rates. Positivity rates were compared between different offers with chi-square tests.

### Profiles of HIV testing users

**User data collection.** Two one-week cross-sectional on-site surveys were planned to determine the characteristics of users who attended the laboratories (through the ALSO programme vs. with a test prescription) and observe changes overtime. To allow comparison with ALSO users, people who sought HIV testing in local STI clinics were asked to participate in the surveys. Community-based HIV testing organisations were not included in the survey given the methods used to reach the most exposed populations and the variety of actions and targeted populations.

All individuals, aged ≥18, seeking an HIV test were invited by the reception staff to answer a self-administered questionnaire in the waiting room before providing a blood sample. Users were informed of the purpose of the study and its voluntary and anonymous nature. Once completed, the questionnaire was placed in an opaque ballot box. Users were informed that completing the questionnaire and placing it in the provided box indicated their consent to participate in the study. Test results were not matched with the questionnaire data, which were anonymously collected.

The questionnaire included 20 items relating to demographic and social characteristics, history of HIV testing, sexual activity and possible HIV exposures in the last 5 years, use of health care in the last 12 months, and reasons for choosing the specific HIV testing facility (i.e., the ALSO programme, prescription, or STI clinic).

The COVID-19 pandemic had a strong impact on mobility and access to laboratories and STI clinics. The pilot study was initially scheduled for 12 months starting in July 2019 but had to be extended until December 2020. The first user survey took place in November 2019 and the second one was postponed in November 2020 during which a second lockdown was implemented. Laboratories experienced high demands related to COVID-19 PCR tests and had to limit occupation of waiting rooms, thus forcing clients to queue outside. Consequently, the survey had to be adapted: while the first one-week survey included all users who received an HIV test at laboratories (ALSO and upon prescription) and at STI clinics, the second survey lasted two consecutive weeks and users tested upon prescription were not asked to participate to lighten the workload of laboratory staff.

**User statistical analyses.** Characteristics of ALSO users were compared to those of people who sought HIV testing in (i) laboratories upon prescription (2019 survey only) and (ii) STI clinics using logistic regression models. Variables associated with ALSO users (p<0.20) were included in a multivariate regression model adjusted for department, age and exposure group (women, heterosexual men, men who have sex with men (MSM)). A stepwise selection procedure was used to select the final model (p<0.05). Analyses were performed using Stata software, release 14.2 (StataCorp LLC, USA).

### Costs of HIV testing offers

The mean costs of HIV testing have been estimated according to negative or positive results of HIV tests carried out as part of the ALSO offer, or upon medical prescription in laboratories, in STI clinics and in community-based organisations (operating on their premises or outdoors

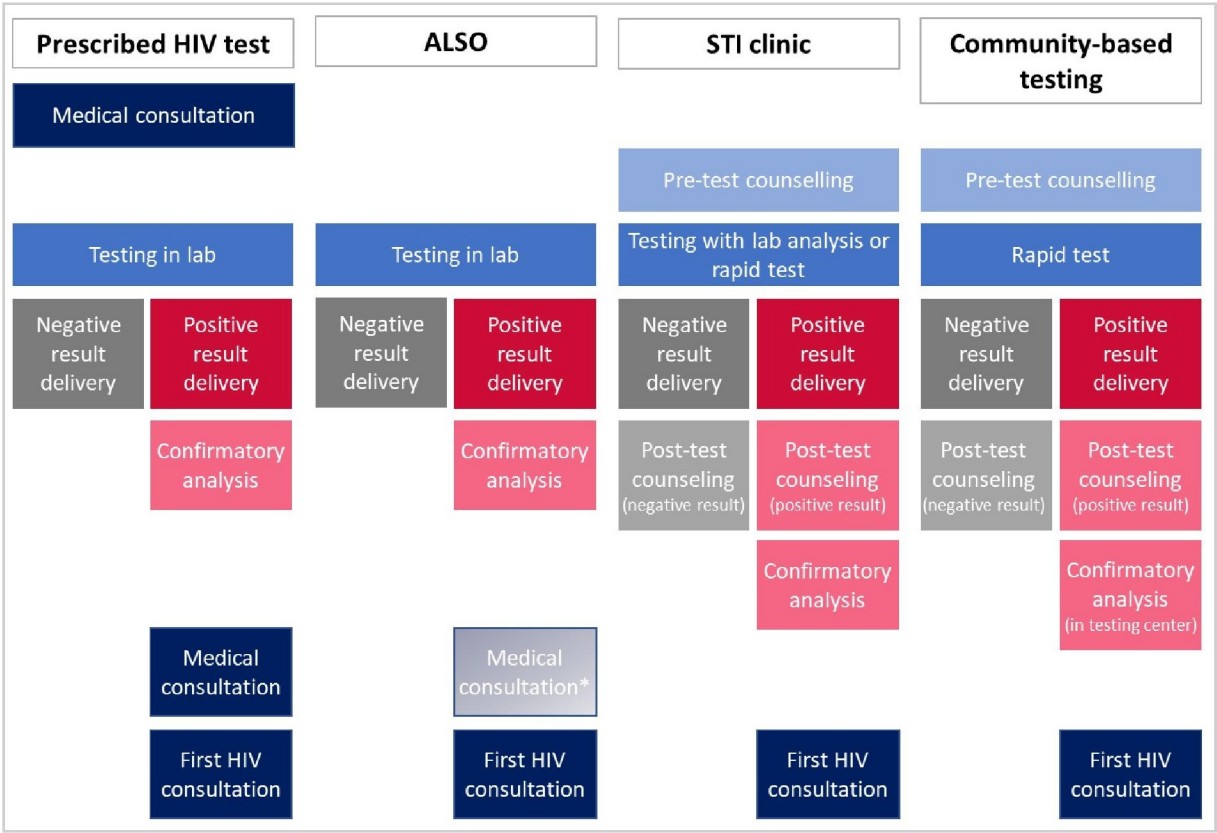

**Fig 1. HIV testing steps considered in estimating mean costs for each testing offer and by test result.** Lab: laboratory. * In the ALSO offer, the biologist delivers a positive result. However, we observed that in one out of three positive tests, the GP was involved in delivering HIV-positive test results. Laboratory analyses were performed according to the national algorithm of HIV diagnosis.

in various community venues). HIV tests performed at the hospital were not considered because data from biological analyses and care are aggregated. Additionally, HIV self-tests were not considered due to lack of data about the actual use after purchase and positivity rates.

**The microcosting approach.** A microcosting approach was applied to each HIV testing offer to estimate the costs as recommended by the French guidelines for the evaluation of a new intervention [19] and by the CDC for HIV testing programmes [20]. We enumerated all resources used and valued them by standardised unit costs. A societal perspective was adopted, i.e., all resources were included regardless of the payer. Only direct costs were included in the analysis: staff, small equipment, reagents and services needed to perform the HIV test [19]. For each offer, as HIV testing is based on an existing system delivering other biological analyses or prevention actions, the costs related to rents, overheads and equipment (fridges, automatons, etc.) were not included. Depending on the case, the prescription of the HIV test or the medical consultations linked to the delivery of the result were also considered.

Fig 1 shows the steps considered in estimating the mean cost for each HIV testing offer. HIV testing analysis was performed in agreement with the national HIV diagnosis algorithm [4]: a first analysis combining ELISA and AgP24 detection; if positive, confirmatory analysis by Western blot on the first sample and ELISA on a second one. The specificity of the combined ELISA/AgP24 measure was 99.8%, and a Western blot was counted for 0.2% of negative results. A positive rapid test should be confirmed by a combined ELISA/AgP24 detection and

a Western blot on one blood sample. When HIV diagnosis was confirmed, costs were counted until the first consultation in a specialised HIV unit [21].

**Cost data collection.** We performed the microcosting approach described above by collecting data from a sample of centres in the two departments where the ALSO pilot was conducted. To assure diversity of studied centres, their selection was stratified on the type of centres, then on the size for laboratory groups (most laboratories are organised into groups to pool resources and analysis equipment) or the attachment to a hospital for STI clinics. Data collection was restricted to July to December 2019, the first semester of the ALSO pilot study without disruptions linked to the COVID-19 pandemic.

A member of the research team collected data on site from staff using a standardised form adapted for each offer. Data were:

- The type of staff and the time in minutes that they were involved in each step for one HIV testing

- Number of pieces of equipment or quantity of products used for sample collection and analysis (gloves, disinfectant, . . .)

- Analysis process and quantity of reagents used for one blood analysis

- Services needed to perform the HIV test (infectious wasted elimination, sample analysis for STI clinics without laboratory, . . .)

- Equipment used by NGOs, especially for outreach HIV testing activities

- HIV testing activity and total activity to identify the part of HIV in all services, products, etc.

- Number of HIV tests performed and positive tests

Communication costs were excluded from the analysis because, in general in France, HIV testing is promoted through national information campaigns, which rarely focus on a single offer.

The head of each centre gave written informed consent for participating in the study. The study collected only operating data and no individual personal data.

**Resource valuation.** By HIV testing offer, for each resource identified, a mean time spent for the task or a mean quantity of product used for one HIV test, for example, was estimated. It was then valued by standardised costs in 2019 Euros: standardised salary grids by position, mean market price for equipment and services.

From the perspective adopted, work of voluntary staff involved in HIV testing offered by NGOs was not valued. The proportion of volunteers and employed workers involved in different HIV testing steps was estimated, and a sensitivity analysis was performed to value their work, using the salary grids of the NGO employees, at equal positions.

The medical visits were valued using conventional fees from the French general nomenclature for professional acts [22].

**Cost statistical analyses.** All resources identified for carrying out one HIV test were counted. Some resources were shared with different analyses (e.g., disinfectants) or tasks. The ratio of HIV activity to total activity and the number of HIV tests performed were used to estimate the part of these resources allocated to one HIV test. The same calculation was used to quantify the portion of total services allocated to one HIV test, as well as the outdoor HIV testing activity of NGOs over the total outreach activities.

Calculation of the mean costs of HIV testing was the same for each HIV testing offer depending on the appropriate step of testing and the test result (Fig 1). A mean cost per step over centres, weighted by the number of HIV testing performed in participating centres, was

estimated from the costs of each resource required for a test. The total cost per offer and per test result was calculated by summing the costs of each step. The calculation included all the steps from admission to a centre or medical consultation (depending on the offer) to (i) the delivery of the result, for the mean cost per negative test; (ii) the first visit to a specialised HIV unit, for the mean cost per positive test.

Then, mean unit costs were applied to the national HIV testing activity in 2019 (data from the LaboVIH surveillance system (Santé publique France) and the NGOs HIV testing report (Direction Générale de la Santé).

## Ethical approval

The study complied with the Commission Nationale Informatique & Libertés reference methodology 004 (CNIL MR0614181119, Clinical Trials NCT04030689).

## Results

### HIV testing activity

Only data of facilities with a complete information over the study period were included into the analysis: in Alpes-Maritimes, all the laboratories (106) and STI clinics (2/2) while in Paris, 108 of 158 laboratories and 5 out of 11 STI clinics.

Over the 18-month period, 38,945 ALSO tests were performed representing 7.2% (26,859/ 372,030) of all HIV tests performed in laboratories included in Paris and 7.3% (12,086/ 165,188) in Alpes-Maritimes. During the same time period, 36,635 HIV tests were performed in the 7 participating STI clinics.

The monthly numbers of ALSO tests were stable from July to December 2019 with an average of 2,700 tests per month in Paris and 800 in Alpes-Maritimes. From February to May 2020, they dropped sharply due to the first COVID-19 lockdown and did not return to the 2019 level during the second half of 2020 (Fig 2). Prescribed tests and tests in STI clinics followed the same trends, with a moderate decrease for prescribed tests, and a greater one in STI clinics. Consequently, the proportion of ALSO tests out of all tests performed in laboratories decreased over the 3-semester period (8.2%, 7.1% and 6.2%).

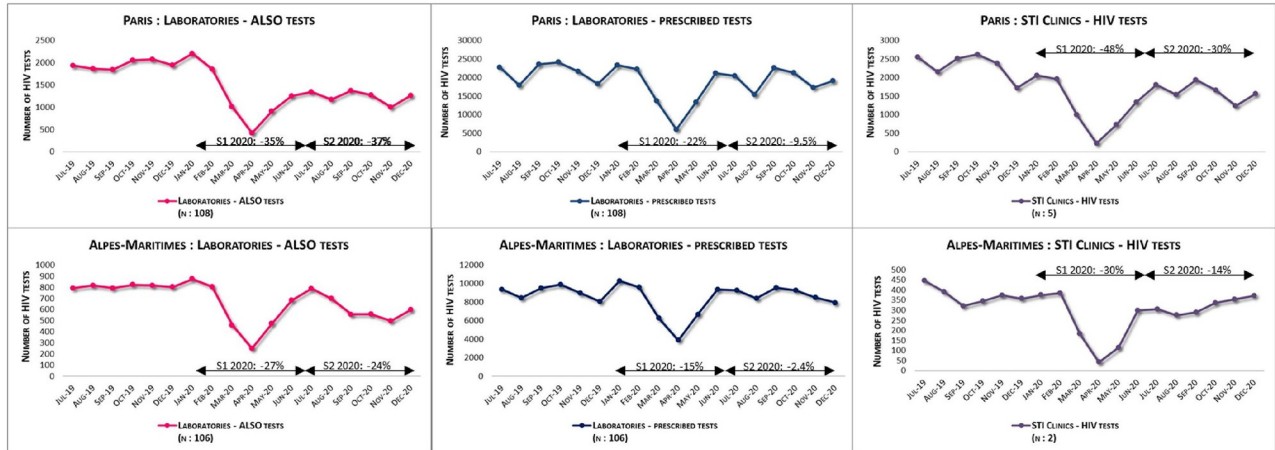

**Fig 2. Monthly number of HIV serologies: ALSO tests and prescribed tests in laboratories and tests at STI clinics.**

**Table 1. Number of ALSO tests and prescribed test in laboratories and number of tests in STI clinics during the pilot study period (July 2019-December 2020).**

| | Laboratories ALSO tests | | | Laboratories Prescribed tests | | | STI clinics | | |
|---|---|---|---|---|---|---|---|---|---|
| | N. tests | + tests: N. | + tests: rate/1000 tests | N. tests | + tests: N. | + tests: rate/1000 tests[c] | N. tests | + tests: N. | + tests: rate/1000 tests[d] |
| Paris[a] | 26,859 | 66 | 2.5 | 345,171 | 752 | 2.2 | 31,072 | 185 | 6.0 |
| Alpes-Maritimes[b] | 12,086 | 19 | 0.9 | 153,102 | 339 | 2.2 | 5,563 | 36 | 6.5 |

[a] 108/158 laboratories and 5/11 STI clinics reported monthly testing activity over the 18 month-period;

[b] 106/106 laboratories and 2/2 STI clinics reported monthly testing activity over the 18 month-period;

[c] ALSO vs. prescribed tests (Chi2): Paris, p = 0.38, Alpes-Maritimes, p = 0.17;

[d] ALSO vs. STI Clinic (Chi2): Paris, p<0.001, Alpes-Maritimes, p<0.001

During the 18-month period, 85 HIV tests were positive among ALSO users, 1,091 among individuals tested on prescription and 221 among STI clinic users. The positivity rates of the ALSO programme did not differ significantly from those of prescribed tests in Paris (2.5 and 2.2 positives/1,000 tests, respectively, p = 0.38) and in Alpes-Maritimes (0.9 and 2.2/1,000 tests, p = 0.17). The positivity rates in STI clinics were significantly higher in both departments (Paris and Alpes-Maritimes), 6.0 and 6.5/1,000 tests, respectively (p<0.001) (Table 1).

Information on the ALSO users with a positive test (n = 85) was only available for the 30 individuals who used the navigation platform to be linked to specialized care. Of them, 18 were MSM, 7 women, and 5 heterosexual men; of these, 6, 5 and 2 respectively, were born abroad. Linkage to care occurred within a median of 5 days. Five (17%) were already aware of their HIV infection but not in care and were relinked to HIV care.

## Profiles of HIV testing users

In November 2019, 125/158 and 87/106 laboratories in Paris and Alpes-Maritimes, respectively, collected at least one questionnaire and; 54/158 and 54/106 laboratories in November 2020. Eleven STI clinics participated in the two surveys (9/11 in Paris, 2/2 in Alpes-Maritimes). Among participants with a valid questionnaire, in 2019, 295 were ALSO users, 2,138 had a prescribed test, and 711 were STI clinic users; in 2020, 388 participants were ALSO users and 573 STI clinic users.

Table 2 presents descriptions and odds ratios resulting from the comparisons of user profiles according to testing offers. Compared to users of prescribed tests, ALSO users were significantly and independently more likely to be a heterosexual man, live outside the department where the laboratory was located, have multiple partners, and have fewer identified HIV exposures in the prior 5 years. They had a worse health insurance coverage (no health insurance or insurance for undocumented migrants) and fewer GP visits. However, they had the same median age of 32 years, and the proportion of first lifetime HIV test was similar (roughly 15%). ALSO users had a lower educational level than users of prescribed tests (54% vs 62% had ≥3 years university level), but this did not remain significant in the multivariate analysis.

Compared to STI clinic users (Table 2), ALSO users were significantly and independently older, more likely to be employed, and live in the same department as the testing location. They had most often a health insurance and had more annual GP visits. Reversely, they were less likely to be exposed to HIV than STI clinic users: having more than 2 partners in the previous year and reporting past HIV exposures were associated with a lower probability of using the ALSO programme rather than visiting an STI clinic. MSM (24% vs 15%), young people (median age: 26 vs 32 years), people born abroad (26% vs 18%) and those who were tested for the first time (25% vs 17%) tended to seek more often HIV testing in STI clinics than in the

**Table 2. Characteristics of users according to the type of testing, on-site survey, November 2019.**

| | | ALSO tests | Prescribed tests | P value ALSO vs. Prescribed test | STI clinics | P value ALSO vs. STI clinics | ALSO tests vs. Prescribed tests | | ALSO tests vs. tests at STI clinics | |
|---|---|---|---|---|---|---|---|---|---|---|
| | | N = 295 | N = 2138 | | N = 711 | | aOR | IC 95% | aOR | IC 95% |
| | | % | % | | % | | | | | |
| Department | Paris | 67.4 | 70.0 | 0.4 | 89.2 | <0.001 | 1 | | 1 | |
| | Alpes-Maritimes | 32.6 | 30.0 | | 10.8 | | 1.12 | 0.82–1.53 | **3.43** | **2.23–3.47** |
| Group | Women | 41.7 | 57.0 | <0.001 | 39.7 | 0.004 | 1 | | 1 | |
| | Heterosexual Men | 42.4 | 28.4 | | 35.0 | | **1.86** | **1.39–2.49** | 1.06 | 0.74–1.52 |
| | MSM | 15.2 | 14.2 | | 24.3 | | 1.53 | 0.99–2.35 | 0.75 | 0.46–1.22 |
| | ND | 0.7 | 0.4 | | 1.0 | | 2.19 | 0.10–12 | 0.70 | 0.11–4.37 |
| Age | median (IQR) | 32 (25–43) | 32 (27–42) | 0.6 | 26 (22–32) | <0.001 | | | | |
| | 16–24 | 20.7 | 17.9 | 0.567 | 42.2 | <0.001 | | | 1 | |
| | 25–30 | 24.1 | 25.0 | | 26.9 | | 1.02 | 0.64–1.64 | 1.58 | 0.94–2.67 |
| | 31–41 | 26.8 | 30.7 | | 18.7 | | 1.00 | 0.61–1.62 | **1.87** | **1.06–3.29** |
| | > = 42 | 27.5 | 25.5 | | 11.5 | | 1.10 | 0.67–1.80 | **3.14** | **1.71–5.75** |
| | ND | 1.0 | 0.9 | | 0.7 | | 1.16 | 0.30–4.47 | 1.94 | 0.28–13.36 |
| Education | <High school | 12.9 | 9.0 | 0.004 | 7.6 | 0.008 | 1.00 | | 1 | |
| | High school | 15.2 | 14.4 | | 21.0 | | 0.77 | 0.47–1.26 | 0.67 | 0.35–1.29 |
| | University (2 years) | 15.9 | 13.4 | | 12.6 | | 0.97 | 0.59–1.59 | 0.95 | 0.49–1.84 |
| | University (> = 3 years) | 53.6 | 61.9 | | 56.4 | | 0.70 | 0.45–1.08 | 0.91 | 0.51–1.60 |
| | ND | 2.4 | 1.3 | | 2.4 | | 1.73 | 0.60–5.01 | 1.28 | 0.34–4.79 |
| Activity | Employed | 70.2 | 72.2 | 0.2 | 47.7 | <0.001 | 1.00 | | 1 | |
| | Unemployed | 9.2 | 7.4 | | 12.5 | | 1.16 | 0.73–1.84 | 0.60 | 0.35–1.02 |
| | Student | 14.2 | 11.0 | | 31.9 | | 1.28 | 0.78–2.09 | **0.55** | **0.32–0.94** |
| | No activity/retired | 5.1 | 6.6 | | 3.9 | | 0.87 | 0.48–1.59 | **0.41** | **0.18–0.91** |
| | ND | 1.3 | 2.8 | | 3.9 | | 0.34 | 0.10–1.13 | **0.25** | **0.07–0.90** |
| Place of living versus place of testing | Same department | 70.5 | 76.0 | 0.013 | 53.0 | <0.001 | 1 | | 1 | |
| | Same region, but not the same department | 18.6 | 13.7 | | 32.8 | | **1.56** | **1.10–2.21** | **0.57** | **0.39–0.84** |
| | Other region in France | 6.4 | 7.7 | | 8.6 | | 0.89 | 0.54–1.49 | 0.64 | 0.34–1.22 |
| | Abroad | 2.7 | 1.0 | | 2.1 | | **2.83** | **1.14–6.98** | 1.55 | 0.56–4.30 |
| | ND | 1.7 | 1.7 | | 3.5 | | 0.87 | 0.30–2.54 | 0.38 | 0.11–1.36 |

*(Continued)*

**Table 2.** (Continued)

| | | ALSO tests | Prescribed tests | P value ALSO vs. Prescribed test | STI clinics | P value ALSO vs. STI clinics | ALSO tests vs. Prescribed tests | | ALSO tests vs. tests at STI clinics | |
|---|---|---|---|---|---|---|---|---|---|---|
| | | N = 295 | N = 2138 | | N = 711 | | aOR | IC 95% | aOR | IC 95% |
| | | % | % | | % | | | | | |
| Place of birth | France | 78.6 | 82.3 | 0.12 | 71.6 | 0.031 | 1.00 | | 1 | |
| | Overseas French regions | 3.4 | 1.9 | | 2.5 | | 1.85 | 0.88–3.88 | 0.94 | 0.37–2.38 |
| | Abroad | 18.0 | 15.6 | | 25.6 | | 1.24 | 0.88–1.75 | 0.68 | 0.45–1.03 |
| | ND | 0 | 0.2 | | 0.4 | | | | | |
| History of testing | Previous test | 81.0 | 84.3 | 0.3 | 73.8 | 0.025 | 1 | | 1 | |
| | First test | 17.0 | 14.5 | | 24.9 | | 1.19 | 0.82–1.73 | 0.67 | 0.43–1.06 |
| | ND | 1.7 | 1.2 | | 1.3 | | 1.71 | 0.57–5.12 | 3.40 | 0.65–17.8 |
| N. of sexual partners over the last 12-month period | 0 or 1 | 28.8 | 42.0 | <0.001 | 15.7 | <0.001 | 1 | | 1 | |
| | > = 2 | 60.0 | 48.6 | | 79.5 | | **1.72** | **1.26–2.34** | **0.45** | **0.30–0.67** |
| | ND | 11.2 | 9.4 | | 4.8 | | 1.28 | 0.76–2.16 | 1.45 | 0.72–2.93 |
| HIV exposures in the previous 5 years | None | 80.7 | 81.0 | 0.034 | 66.9 | <0.001 | 1 | | 1 | |
| | 1 | 16.3 | 12.8 | | 24.9 | | 0.93 | 0.64–1.37 | **0.58** | **0.38–0.90** |
| | > = 2 | 3.1 | 6.2 | | 8.2 | | **0.34** | **0.16–0.73** | **0.26** | **0.12–0.60** |
| Ever feared to be HIV infected | No | 52.9 | 64.1 | <0.001 | 42.2 | 0.008 | **1** | | 1 | |
| | Yes | 39.0 | 29.2 | | 47.1 | | **1.61** | **1.20–2.16** | 0.85 | 0.60–1.21 |
| | ND | 8.1 | 6.7 | | 10.7 | | 1.23 | 0.73–2.07 | **0.47** | **0.60–0.85** |
| Health insurance | Regular insurance | 88.5 | 90.9 | 0.010 | 79.0 | <0.001 | 1 | | 1 | |
| | AME or none | 6.1 | 2.8 | | 14.8 | | **1.84** | **1.03–3.28** | **0.47** | **0.26–0.86** |
| | ND | 5.4 | 6.3 | | 6.2 | | 0.42 | 0.10–1.74 | 0.58 | 0.13–2.55 |
| Visit to GP over the last 12-month period | None | 6.4 | 2.7 | <0.001 | 9.4 | 0.3 | **1.90** | **1.05–3.41** | **0.52** | **0.27–0.97** |
| | 1 visit | 25.1 | 18.7 | | 24.5 | | **1.39** | **1.02–1.89** | 0.85 | 0.58–1.24 |
| | >1 visit | 62.4 | 72.3 | | 59.6 | | 1 | | 1 | |
| | ND | 6.1 | 6.3 | | 6.5 | | 2.19 | 0.56–8,60 | 1.41 | 0.34–5.90 |

ND: missing data or do not wish to answer. AME for "Aide Médicale d'Etat": provisory health insurance offered to undocumented migrants

ALSO programme. However, these differences were not significant in the multivariate analysis.

Participants in the 2019 survey had to choose the two main reasons why they selected the specific HIV testing programme from an 11-item list. ALSO users selected proximity to their home (64%), and lack of appointments (31%) likewise prescribed tests users (71% and 20%, respectively). STI clinics were chosen because they were free (63%) and close to home (47%).

**Table 3. Estimated costs of HIV testing.** a) Estimated mean costs of HIV testing offers according to test results. b) estimation of HIV testing activity and costs at the national level in 2019 if the ALSO offer was extended to all of France.

| | Prescribed test | ALSO | STI clinic | Community-based testing |
|---|---|---|---|---|
| **a) Mean cost (in €) of** | | | | |
| One negative HIV test | 37.90 | 12.57 | 33.72 | 40.31 |
| One positive HIV test | 216.09 | 162.86 | 169.33 | 190.90 |
| **b) Extrapolation of HIV testing activity and costs at the national level in 2019** | | | | |
| Number of HIV tests | 3,953,895 | 350,784 | 397,868 | 69,356 |
| Number of HIV-positive tests* | 7,512 | 842 | 1,531 | 555 |
| Total cost (€) | 151,191,267 | 4,535,877 | 13,879,811 | 2,879,295 |
| Cost to identify one positive result (€) | 20,126 | 5,388 | 9,068 | 5,189 |

\* The positive rate was 1.9‰ for prescribed tests [9], 2.4‰ for the ALSO test, and 8.0‰ for community-based testing (Direction Générale de la Santé). In STI clinics, the positivity rate was 3.5‰ for rapid testing and 3.9‰ for lab testing. At the national level, 13.2% of HIV tests in testing centres were rapid tests (Santé publique France).

In 2020, most of the differences observed in 2019 between ALSO and STI clinic users were found (S1 Table). Among ALSO participants in the 2020 survey, 19% reported an ALSO HIV test in the previous 12 months. These repeated ALSO users were more likely to be heterosexual men (46% vs. 28% for repeated users who were tested on prescription or attended an STI clinic, p<0.0001).

## Cost of HIV testing offers

The participating centres to the cost study were four laboratory groups of different sizes that performed 45,520 HIV tests upon prescription and 5,268 tests in ALSO offer (ALSO tests) from July to December 2019; four STI clinics attached or not to a hospital (7,056 HIV tests); two community-based organisations that deliver voluntary counselling and testing (one local and one national, 17,268 HIV tests).

The mean costs per positive test and per negative test are presented in Table 3 (and by HIV testing step (Fig 1) in the S2–S6 Tables). Regardless of the HIV testing offer, the main differences in costs by test result were related to confirmatory analyses and longer time spent by the provider to deliver the positive result. For an HIV test carried out in a laboratory upon medical prescription, the mean cost was €37.90 for a negative test and €216.09 for a positive test (Table 3 and S2 Table).

In the ALSO programme, laboratory costs were close to those of prescribed tests (without a medical prescription, S3 Table). Most positive results were delivered by the biologist face-to-face. We observed, in one third of cases, that the GP was required either by the biologist or the patient to support the delivery of the positive test. The mean cost of the ALSO offer was €12.57 for a negative test and €162.86 for a positive test.

In STI clinics, two types of HIV tests can be performed: tests on blood samples sent to laboratories for analysis and rapid tests on fingertip prick capillary blood (that represented 16.9% of total HIV tests in studied STI clinics). A large part of the time spent on a test was for pre- and post-test counselling. Compared to a conventional test, the cost of a rapid test was lower (roughly €19 versus €37 for a negative result) because it was more likely performed by nurses rather than physicians (S4 and S5 Tables). Moreover, a psychologist helped in delivering one out of two positive test results. The mean cost of HIV testing in STI clinics was €33.72 for a negative test and €169.33 for a positive test.

In community-based testing, community workers used only rapid HIV tests (S6 Table). Compared to HIV testing in STI clinics, the mean durations of counselling were longer (40 to 90 minutes in NGOs versus 24 to 40 minutes in STI clinics, depending on the test result). Volunteers were more likely to do testing proposals and admissions (80% of testing), while employed community workers did counselling and rapid testing (80% of testing). When a rapid test was positive, the tested person was referred to an STI clinic for a confirmatory analysis and then directly to an HIV specialist. The mean cost of community-based HIV testing was €40.41 for a negative test and €190.90 for a positive test. In valuing volunteer work, the mean cost for an HIV test increased from €40.41 to €68.44 when the test was negative and from €190.90€ to €222.50 when the test was positive. Volunteering would represent an additional cost of approximately €30 per test, regardless of the test result.

We have estimated costs of the ALSO programme as if it was extended to all of France in 2019. We assumed the proportion of tests performed and positivity rate were similar to those of the two French departments where the offer was implemented (8.1% of HIV testing activity in laboratories; 2.4‰) and throughout the one-year period. Considering the four testing offers, approximately 4.8 million tests would have been performed in 2019 for a total cost of €172 million. With the ALSO offer, 842 HIV-positive people would have been identified for a total cost of €4.5 million. The cost of the ALSO offer represented 2.6% of the overall cost of HIV testing. The cost to identify an HIV-positive person was €5,388 through the ASLO offer. This cost was close to the cost of community-based testing (€5,189) and much lower than the cost of STI clinics and prescribed tests (€9,068 and €20,126, respectively; Table 3).

## Discussion

The ALSO programme provided access to HIV testing at laboratories, without requiring a prescription, payment, or appointment. This programme led to a net increase in HIV testing in laboratories, with 7% of HIV testing volume attributed to the ALSO offer. The positivity rate was similar to that of prescribed tests, however, costs for performing the HIV test were the lowest.

The COVID-19 pandemic prevented from assessing the trends overtime of ALSO use in optimal conditions, especially to identify possible shifts to ALSO from pre-existing offers. In 2020, the total volume of HIV testing fell nationwide. In both departments, the volume of ALSO tests fell more than that of prescribed tests. However, although laboratory access worsened (outdoor queues, important COVID testing activity) and communication campaigns were halted, the proportion of ALSO tests remained above 6% in the second half of 2020.

ALSO users had characteristics between those of individuals who used existing HIV testing offers in terms of age, sexual exposure, health insurance coverage, use of care, and geographic proximity. A key finding was that the ALSO programme reached heterosexual men with >2 partners a year. These men are underrepresented in other testing offers while their estimated time between HIV infection and diagnosis was of more than 4 years [10]. The underlying hypothesis is that HIV testing at laboratories may be more comfortable for them, without the need to talk about their sexual life. In addition, easing accessibility may facilitate the transition from intention to action. These factors may also favour repeated testing. In the 2020 survey, one in five ALSO users had already used the ALSO programme for a previous test, suggesting that this testing may contribute to habit formation, especially among heterosexual men. This should be however evaluated in the long term.

The ALSO programme did not reach significantly more first-time testers or migrants. Younger people appeared to use preferentially STI clinics due to their specialisation, and renown throughout population. Migrants preferred STI clinics too and are effectively targeted

by community-outreach programmes [23]. However, although information on characteristics of people who received a positive test result were limited by the small number of individuals who sought the ALSO navigation support, 40% of them were born abroad. Other innovative programmes are needed to reach the most recently arrived migrants, vulnerable to HIV due to hard living conditions in months following arrival [24].

ALSO costs were low and represented a very small part of the overall cost of HIV testing (2.6%). HIV testing costs depend on (1) who carried out the test (physician, nurse or community-worker); (2) the time spent by staff in each HIV step (reach client / propose the test, pre- and post-test counselling) [25], and (3) the proportion of HIV infected people among the users. NGOs operate in or near venues attended by key populations. Their users are highly exposed to HIV, with >60% of them belonging to a high-risk group, highlighted by the highest HIV positivity rate (8/1,000) [23, 26]. Consequently, the cost of finding an HIV-positive person is the lowest. HIV testing is offered as comprehensive prevention information. The time spent referring people for testing and counselling represents added value and higher costs [5, 6]. In contrast, the ALSO programme, without counseling, involving little or no physician, had the lowest cost per negative and positive test and a cost of finding an HIV-positive person with the ALSO offer of €5,388, close to the cost of community-based testing. Those who get tested upon medical prescription were the least exposed to HIV. Although the positivity rate was the lowest (1.9/1,000), testing in the laboratory upon prescription found the highest number of HIV infections, given the high number of tests performed each year [9].

To our knowledge, there is no HIV testing offer similar to ALSO published in the literature; however, the implementation of this type of programmes depends on the organization and accessibility of the healthcare system, and in particular, the different testing provisions, of each country. In France, laboratories are places of proximity with free access for people whatever the medical analysis they must be carried out. This makes them places of choice for evaluating a new HIV testing offer. The ALSO programme might be considered as an alternative to self-testing that removes some geographical and psychological barriers [15, 27–29]. Self-test sales remained limited in high-income countries, including France, mainly because of price and poorer confidence in HIV self-testing [28, 30–33]. The ALSO programme and self-testing both lack in-person counselling, which might meet the preferences of those tested frequently. ALSO users highlighted proximity and convenience in choosing the ALSO offer. The characteristics of the HIV testing offers (accessibility, proximity, testing through medical provider or not, counselling, confidentiality) impact on their attractiveness and populations reached, showing their complementarity.

The strength of this study was that it evaluates a new HIV testing offer in terms of activity, users and costs. It represents the first time that users who seek HIV testing at laboratories, either with a prescription or through the ALSO programme, and in STI clinics were surveyed during the same periods, at the time of testing, and with the same questionnaire. In France, to date, HIV testing history and associated determinants have been only documented retrospectively from population surveys [34, 35]. Another strength was the cost evaluation of the programme and of the other three main HIV testing offers. The microcosting approach used is the most comprehensive and precise method of estimating the costs of an intervention [12]. Even if the estimated costs did not reflect exactly the actual costs of testing (they were different from the reimbursement rate by French health insurance or the subsidies received by STI clinics or NGOs), the strength of our approach is that we estimated the costs in the same way for each offer. It highlighted the resources used for testing in particular those that were used more in one offer than in another [8]. Many studies have compared the cost-effectiveness of HIV testing programmes or strategies with mathematical models using testing coverage and diagnoses as measures of effectiveness [36–38]. Our study, carried out in real-life conditions, with

cost data observed on a representative sample of centres, enabled us to also consider the profiles of the populations reached by each offer at the same time [39, 40].

The study has some limitations. As already addressed above, the crisis due to the COVID-19 pandemic disturbed the implementation of the experimentation and the data collection. Laboratories were overwhelmed by the COVID testing and several of them did not participate in the second cross-sectional survey. To estimate the selection bias that could result, the characteristics of ALSO participants of the 2019 cross-sectional survey were compared according to the participation of the laboratory in one or two surveys. More Parisian laboratories did not participate in the second surveys that implied that participants were less likely to be born abroad or live in another department than those of the testing (data not shown). No other differences were highlighted suggesting a limited selection bias. Another limitation is that we have no information about linkage to care for 65% of the patients who received a positive ALSO test result. In order to respect the confidentiality of the tested people, we were not able to collect this data for the people who chose not to use the navigation platform.

## Conclusion

The evaluation of a new testing programme considering other HIV testing offers highlighted the effectiveness of the ALSO program in terms of activity, HIV exposed populations reached and costs. Besides provider-initiated testing, dedicated STI clinics and community-based testing targeting the most exposed groups, the ALSO pilot study creates complementary opportunities to increase the spontaneous demand for testing in the primary care services, i.e. laboratories and by relying on universal health coverage. In the French context, ALSO could be an additional testing strategy needed to achieve the 3x95 target. [41] French health authorities decided to expand ALSO nationally from 2022 [42]. Challenges to national implementation include large communication campaigns and ensuring linkage to care for newly diagnosed users based on dedicated navigation platforms.

## Supporting information

**S1 Table. Characteristics of users according to the type of testing, ALSO or STI clinics, on-site survey, November 2019 and November 2020.**
(DOCX)

**S2 Table. Mean costs of HIV testing, by step and in total, according to test results, estimated by microcosting for an HIV test carried out in a laboratory (lab) after a medical prescription (prescribed test, PT).**
(DOCX)

**S3 Table. Mean costs of HIV testing, by step and in total, according to test results, estimated by microcosting for an ALSO test carried out in laboratory (lab).**
(DOCX)

**S4 Table. Mean costs of HIV testing, by step and in total, according to test results, estimated by microcosting for an HIV test carried out in STI clinic.**
(DOCX)

**S5 Table. Mean costs of HIV testing, by step and in total, according to test results, estimated by microcosting for a rapid HIV test carried out in STI clinic.**
(DOCX)

**S6 Table. Mean costs of HIV testing, by step and in total, according to test results, estimated by microcosting for a rapid HIV test carried out in community-based organisation**

**(CBO).**
(DOCX)

## Acknowledgments

The authors thank all laboratory staff who implemented the ALSO programme and participated in all the parts of data collection, teams at sexually transmitted infection (STI) clinics who implemented the user survey, and the users who agreed to participate in the survey.

The authors thank the persons from the PACA East and the four Ile-de-France Corevihs who were involved in the navigation platform: Christophe Caissotti, Morgane Marcou, Laurent Richier, Agnès Cros, Carole Louisin, Marie-Pierre Pietri, Valérie Le Baut, Claude Mackoumbou-Nkouka, Gersende Grain, Zélie Julia, Françoise Louni, Cindy Godard, Malikhone Chansombat, Awa Ndiaye, Stéphanie Cossec, Mouniya Mebarki, Anne Adda-Lievin, Nadir Gaad, Pélagie Thibaut, Céline Wilpotte, Manuela Sébire, Julie Lamarque, Christian Thanh Huy Tran, Naoual Qatib, Yasmine Dudoit, Christine Blanc, Ludovic Lenclume, Dalila Beniken.

Data collection in Paris was conducted by Clinsearch; in Alpes-Maritimes, data collection was conducted by the clinical research unit of the Nice teaching hospital.

ALSO group:

Laurence Dauffy, Laurence Dumondin, Anne-Claire Haye (Health Insurance Paris); Gwenaëlle Tasset, Sarah Coquillat, Gérard Ughetto (Health Insurance, Alpes-Maritimes); Florence Orsini, Saïd Oumeddour (National Health Insurance); Jean-Claude Azoulay (URPS Biology, Île-de-France); Boris Loquet (URPS Biology Provence Alpes-Côte d'Azur (PACA)); Frédéric Goyet, Corinne Chouraqui (Île-de-France Region Health Agency), Isabelle Virem (PACA Region Health Agency), Anne Souyris (City of Paris), Marion Vandenbrouck (City of Nice).

Lead author for the ALSO group: Pascal Pugliese, pugliese.p@chu-nice.fr

## Author Contributions

**Conceptualization:** Karen Champenois, Victoire Sawras, Nathalie Lydié, Elodie Aïna, Erwann Le Hô, Eve Plenel, Sylvie Deuffic-Burban, France Lert, Pascal Pugliese.

**Data curation:** Victoire Sawras, Pamela Ngoh, Philippe Bouvet de la Maisonneuve.

**Formal analysis:** Karen Champenois, Victoire Sawras, Philippe Bouvet de la Maisonneuve, Margot Annequin, France Lert.

**Funding acquisition:** Karen Champenois, Eve Plenel, France Lert, Pascal Pugliese.

**Investigation:** Pamela Ngoh, Philippe Bouvet de la Maisonneuve, Julie Valbousquet, Yoana Gatseva, David Michels, Charlotte Maguet, Irit Touitou.

**Methodology:** Karen Champenois, Margot Annequin, Nathalie Lydié, Sylvie Deuffic-Burban, France Lert.

**Project administration:** Pamela Ngoh, Julie Valbousquet, Charlotte Maguet, Elodie Aïna, Erwann Le Hô, Eve Plenel.

**Supervision:** Julie Valbousquet, Elodie Aïna, Eve Plenel, France Lert, Pascal Pugliese.

**Validation:** Pascal Pugliese.

**Writing – original draft:** Karen Champenois, Victoire Sawras, France Lert.

**Writing – review & editing:** Pamela Ngoh, Philippe Bouvet de la Maisonneuve, Julie Valbousquet, Margot Annequin, Yoana Gatseva, David Michels, Nathalie Lydié, Charlotte Maguet,

Elodie Aïna, Erwann Le Hô, Eve Plenel, Irit Touitou, Sylvie Deuffic-Burban, Pascal Pugliese.

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
