## [Decision Letter · Decision Letter 0]

18 Mar 2024

PONE-D-23-11875Facilitating the access to HIV testing at lower costs: “To the laboratory without prescription” (ALSO), a pilot intervention to expand HIV testing through medical laboratories in FrancePLOS ONE

Dear Dr. Champenois,

Thank you for submitting your manuscript to PLOS ONE. After careful consideration, we feel that it has merit but does not fully meet PLOS ONE’s publication criteria as it currently stands. Therefore, we invite you to submit a revised version of the manuscript that addresses the points raised during the review process.

We look forward to receiving your revised manuscript.

Kind regards,

Prakash Shakya

Academic Editor

PLOS ONE

Journal Requirements:

“The ALSO study was supported by Santé publique France (no grant number, https://www.santepubliquefrance.fr) and ANRS MIE (ECTZ118440, https://www.anrs.fr/). The organisations Vers Paris sans Sida, Objectif sida zero and Inserm received the funds. No author has directly received the funds.

3. In this instance it seems there may be acceptable restrictions in place that prevent the public sharing of your minimal data. However, in line with our goal of ensuring long-term data availability to all interested researchers, PLOS’ Data Policy states that authors cannot be the sole named individuals responsible for ensuring data access (http://journals.plos.org/plosone/s/data-availability#loc-acceptable-data-sharing-methods).

4. One of the noted authors is a group or consortium [ALSO group]. In addition to naming the author group, please list the individual authors and affiliations within this group in the acknowledgments section of your manuscript. Please also indicate clearly a lead author for this group along with a contact email address.

Reviewers' comments:

Reviewer's Responses to Questions

**Comments to the Author**

1. Is the manuscript technically sound, and do the data support the conclusions?

Reviewer #1: Yes

Reviewer #2: Yes

2. Has the statistical analysis been performed appropriately and rigorously? 

Reviewer #1: Yes

Reviewer #2: I Don't Know

3. Have the authors made all data underlying the findings in their manuscript fully available?

Reviewer #1: Yes

Reviewer #2: No

4. Is the manuscript presented in an intelligible fashion and written in standard English?

Reviewer #1: Yes

Reviewer #2: Yes

5. Review Comments to the Author

Reviewer #1: Excellent job. My only suggestions are: (1) describe (in methodology, page 5) how long it took to get the results after testing (at ALSO testing sites); and (2) on page 11, in results (lines 251 and 252), give any significant differences between the Paris labs (108/158) that performed ALSO vs those that did not, and the STI clinics (5/11).

Reviewer #2: General comments:

This was a study assessing the impact of a program named « Au Labo sans ordo (ALSO) - To the laboratory without prescription » performed in two major French cities, Paris and Nice, which have the highest annual rate of HIV diagnoses and prevalence of undiagnosed cases. The program consisted in allowing patients to go to any laboratory during opening hours without a prescription and asking for free HIV testing (laboratories were reimbursed by the national health insurance), without restriction, without appointment, and without pre-test counseling.

They found that about 7% of HIV test were performed through the ASLO program during the study period and that the population attending this program was not the same as in STI clinics or as in prescribed HIV testing as it reached in particular heterosexual men with more than 2 partners a year, highlighting the complementarity of the program in HIV testing offer. The positivity rate was similar as in prescription test but lower than in STI clinics. Moreover, they found that the cost per HIV test was the lowest in ALSO and that the costs to obtain one positive test were lower than for prescribed test and in STI clinic and close from community-based testing where the HIV prevalence is the highest.

The article is clear, well written and well discussed. The cost analysis is a significant addition to the study. The study and the program are of public health interest and offer perspectives in improving access to HIV testing. The present manuscript will be of interest for the Plos One readers.

I have a few comments

Major:

- Did this program concerned only adults ? I guess that yess but it should be indicated.

- Do the authors have any information about how many positive patients effectively went to their GP or to the dedicated navigation platform then to an HIV clinic ?

- Should a multinomial logistic regression be performed rather than two different binomial logistic regressions (also vs. prescribed and also vs. STI)?

- Does the authors have any information about redundancy in HIV testing (e.g. patients using ALSO and STI clinic)?

- Is there in the literature any similar approach as the ALSO program? If no, it should be stated. If yes, programs and results should be compared.

- Authors should highlight the main limitations of their study alongside the strengths (impact of the COVID-19 pandemic on the evaluation for the programme, less laboratories collecting a questionnaire in the second study period, )

Minor:

- Line 31: indicate in which year 51% of newly diagnosed HIV patients had CD4<350

- Line 37: indicate that tests can be anonymous

6. PLOS authors have the option to publish the peer review history of their article (what does this mean?). If published, this will include your full peer review and any attached files.

Reviewer #1: **Yes: **John Nelson, PhD, CNS, CPNP

Reviewer #2: **Yes: **Maxime Hentzien

---

## [Author Response · Author response to Decision Letter 0]

31 May 2024

Facilitating the access to HIV testing at lower costs: “To the laboratory without prescription” (ALSO), a pilot intervention to expand HIV testing through medical laboratories in France

Authors ‘response to comments

Review Comments to the Author

We thank the reviewers for their careful reviewing of our manuscript and their valuable comments that helped us to improve the manuscript. You will find below a point by point response to reviewers’ comments.

Reviewer #1

Excellent job. My only suggestions are: 

(1) describe (in methodology, page 5) how long it took to get the results after testing (at ALSO testing sites); 

In labs, regardless of the ALSO offer or prescribed test, people can obtain the results on the same day as the sample or the next day if they come at the end of the day. When the result is negative, they access to it online. When it is positive, the biologist called the patient by phone to invite him/her to come to the laboratory for the confirmatory analysis on a second blood sample.

We cleared the text as follows:

L94-95: “Results were generally available the same day.”

(2) on page 11, in results (lines 251 and 252), give any significant differences between the Paris labs (108/158) that performed ALSO vs those that did not, and the STI clinics (5/11).

Offering ALSO was not optional for laboratories in Paris and the Alpes Maritimes since the program was decided by the Health Insurance and then part of the regular provision of HIV testing. In Paris, only 68% of the laboratories reported monthly HIV testing data during the whole 18 month-period. Very few of them did not report any information. 

We compared the number of ALSO tests reported by laboratories to the aggregated number of tests reimbursed by health insurance (comparisons were possible on aggregated data only). The numbers were a little different but trends were similar which makes us confident in our results.

6 of the 11 Parisian STI clinics did not provide activity data because of important disruptions of their organisation due to the covid crisis.

Reviewer #2

General comments:

This was a study assessing the impact of a program named « Au Labo sans ordo (ALSO) - To the laboratory without prescription » performed in two major French cities, Paris and Nice, which have the highest annual rate of HIV diagnoses and prevalence of undiagnosed cases. The program consisted in allowing patients to go to any laboratory during opening hours without a prescription and asking for free HIV testing (laboratories were reimbursed by the national health insurance), without restriction, without appointment, and without pre-test counseling.

They found that about 7% of HIV test were performed through the ASLO program during the study period and that the population attending this program was not the same as in STI clinics or as in prescribed HIV testing as it reached in particular heterosexual men with more than 2 partners a year, highlighting the complementarity of the program in HIV testing offer. The positivity rate was similar as in prescription test but lower than in STI clinics. Moreover, they found that the cost per HIV test was the lowest in ALSO and that the costs to obtain one positive test were lower than for prescribed test and in STI clinic and close from community-based testing where the HIV prevalence is the highest.

The article is clear, well written and well discussed. The cost analysis is a significant addition to the study. The study and the program are of public health interest and offer perspectives in improving access to HIV testing. The present manuscript will be of interest for the Plos One readers.

I have a few comments

Major:

1) Did this program concerned only adults? I guess that yes but it should be indicated.

In France, HIV testing is available to minors aged 15 or over without parental consent but in the presence of a trusted adult chosen by the minor (the trusted adult can be a relative, a friend or a member of testing staff; Articles L.1111-5 et L1111-5-1 du Code de la santé publique). However, the exemption from parental authority does not apply to the biologist, the ALSO offer is available to people aged from 18 years old. 

This has been added in the manuscript as follows:

L88: “The ALSO programme was open to all potential users in France, aged 18 or over.”

2) Do the authors have any information about how many positive patients effectively went to their GP or to the dedicated navigation platform then to an HIV clinic?

As described lines 283-87, 85 ALSO users received a positive result, only 30 of them wished to use the navigation platform. Navigation does not yet exist in France for HIV testing and was offered as part of the ALSO program. Usually patients with a positive test are referred to the physician who issued the prescription or make an appointment directly at the hospital HIV clinic of their choice. Some laboratory staffs, most often in Paris where HIV epidemic is the highest in France, reported not using the navigation platform because they used to refer patient who tested positive for HIV to the nearest HIV clinic. Unfortunately, we have no information on actual linkage to GP or HIV care for these positive individuals. 

We added this information as a limitation in the discussion part (lines 470-73)

Another limitation is that we have no information about linkage to care for 65% of the patients who received a positive ALSO test result. In order to respect the confidentiality of the tested people, we were not able to collect this data for the people who chose not to use the navigation platform. 

3) Should a multinomial logistic regression be performed rather than two different binomial logistic regressions (also vs. prescribed and also vs. STI)?

The estimates of several logistic regressions or a multinomial logistic regression are the same, with the difference that the standard errors are a little larger in the case of several logistic regressions [Agresti A. Logit Models for Multinomial Responses. Categorical Data Analysis. John Wiley & Sons, Ltd; 2002. pp. 267–313. doi:10.1002/0471249688.ch7]. 

We chose to do separate logistic models despite the increase in standard errors, because:

- We didn’t want to compare people with a prescribed test and STI clinic users who we knew they were different

- We wanted to express the probability of using HIV testing in comparison to other screening methods (HIV testing on prescription or in STI clinics are and will remain the norm).

In presenting the results of a multinomial logistic regression, we were obliged to put ALSO users as a reference category and therefore to present the probability of using prescribed tests or an STIs clinics compared to ALSO users. 

For the reviewer’s information, the results of the multinomial analysis are presented at the end of the attached file “ReviewerCommentAnswer” .

4) Does the authors have any information about redundancy in HIV testing (e.g. patients using ALSO and STI clinic)?

We did not have exactly these data. However, with the cross-sectional study to characterize the HIV testing users, we can know the number of HIV tests carried out in the previous 12 months and the HIV testing facility of the last HIV test.

The table below presented these data by HIV testing offer. The majority of users used the same HIV testing offer for both tests, otherwise the laboratory with a medical prescription.

 ALSO offer Prescribed test STI clinic

Number of users who stated having carried out ≥2 HIV tests in the last 12 months

(% of the total users) 30 (10.2%) 277 (13.0%) 103 (14.5%)

Facility of the last HIV testing 

In a lab, with a medical prescription 8 (32%) 231 (88%) 23 (24%)

In a lab, without a medical prescription 13 (52%)* 13 (5%) 9 (10%)**

STI clinic 4 (16%) 15 (5%) 55 (59%) 

Other facility 0 5 (2%) 7 (7%)

Missing data 5 13 9

* Among the 13 users who stated having carried out their last HIV test in a laboratory without a medical prescription, 11 had taken their last HIV test in the 6 prior months so it might already be an ALSO test.

** Among the 9 users who stated having carried out their last HIV test in a laboratory without a medical prescription, 7 had taken their last HIV test in the 6 prior months so it might already be an ALSO test.

We chose not to add these results to the manuscript because there remains some uncertainty due to the small number of respondents and a possible recall bias.

5) Is there in the literature any similar approach as the ALSO program? If no, it should be stated. If yes, programs and results should be compared.

To our knowledge, no similar approach to the ALSO offer was published. This type of offer depends on the organization of the healthcare system and the accessibility of the different testing provisions in each single country.

We added this paragraph in the discussion part (lines 431-5)

To our knowledge, there is no HIV testing offer similar to ALSO published in the literature; however, the implementation of this type of programmes depends on the organization and accessibility of the healthcare system, and in particular, the different testing provisions, of each country. In France, laboratories are places of proximity with free access for people whatever the medical analysis they must be carried out. This makes them places of choice for evaluating a new HIV testing offer.

6) Authors should highlight the main limitations of their study alongside the strengths (impact of the COVID-19 pandemic on the evaluation for the programme, less laboratories collecting a questionnaire in the second study period, ) 

We thank the reviewer for highlighting this point. We added a paragraph addressing the limitations at the end of the discussion part (lines 462-73)

The study has some limitations. As already addressed above, the crisis due to the COVID-19 pandemic disturbed the implementation of the experimentation and the data collection. Laboratories were overwhelmed by the COVID testing and several of them did not participate in the second cross-sectional survey. To estimate the selection bias that could result, the characteristics of ALSO participants of the 2019 cross-sectional survey were compared according to the participation of the laboratory in one or two surveys. More Parisian laboratories did not participate in the second surveys that implied that participants were less likely to be born abroad or live in another department than those of the testing. No other differences were highlighted suggesting a limited selection bias. Another limitation is that we have no information about linkage to care for 65% of the patients who received a positive ALSO test result. In order to respect the confidentiality of the tested people, we were not able to collect this data for the people who chose not to use the navigation platform. 

Minor:

7) Line 31: indicate in which year 51% of newly diagnosed HIV patients had CD4<350

According to the 2020 ECDC data, 51% of diagnoses with CD4 available CD4 counts were late. We did not update this data despite availability of 2022 data because, it is related the study period.

We added this information in the text (line 31).

8) Line 37: indicate that tests can be anonymous

Done 

Journal Requirements

The manuscript was edited as recommended. Corresponding changes (changes in statements position or the police of the titles) were not highlighted in the revised version.

“The ALSO study was supported by Santé publique France (no grant number, https://www.santepubliquefrance.fr) and ANRS MIE (ECTZ118440, https://www.anrs.fr/). The organisations Vers Paris sans Sida, Objectif sida zero and Inserm received the funds. No author has directly received the funds. 

The funding statement was edited as recommended.

3. In this instance it seems there may be acceptable restrictions in place that prevent the public sharing of your minimal data. However, in line with our goal of ensuring long-term data availability to all interested researchers, PLOS’ Data Policy states that authors cannot be the sole named individuals responsible for ensuring data access (http://journals.plos.org/plosone/s/data-availability#loc-acceptable-data-sharing-methods).

Data collected cannot be shared publicly because of their sensitive nature and permission for public use of the data was not obtained from the participants. Data are stored on a secure server suitable for hosting health data. Activity and user data are available upon request to the corresponding author or to the data protection officer of the Nice hospital: dpo@chu-nice.fr for researchers who meet the criteria for access to confidential data, after receiving an approval from the ALSO study group. 

This statement was added in the manuscript lines 533-540.

4. One of the noted authors is a group or consortium [ALSO group]. In addition to naming the author group, please list the individual authors and affiliations within this group in the acknowledgments section of your manuscript. Please also indicate clearly a lead author for this group along with a contact email address.

The lead author for the ALSO group is Pascal Pugliese, pugliese.p@chu-nice.fr

This has been added in the acknowlegment section.

The references have been reviewed carefully.

---

## [Decision Letter · Decision Letter 1]

19 Aug 2024

Facilitating the access to HIV testing at lower costs: “To the laboratory without prescription” (ALSO), a pilot intervention to expand HIV testing through medical laboratories in France

PONE-D-23-11875R1

Dear Dr. Champenois,

We’re pleased to inform you that your manuscript has been judged scientifically suitable for publication and will be formally accepted for publication once it meets all outstanding technical requirements.

Kind regards,

Prakash Shakya

Academic Editor

PLOS ONE

Additional Editor Comments (optional):

Reviewers' comments:

Reviewer's Responses to Questions

**Comments to the Author**

1. If the authors have adequately addressed your comments raised in a previous round of review and you feel that this manuscript is now acceptable for publication, you may indicate that here to bypass the “Comments to the Author” section, enter your conflict of interest statement in the “Confidential to Editor” section, and submit your "Accept" recommendation.

Reviewer #2: All comments have been addressed

2. Is the manuscript technically sound, and do the data support the conclusions?

Reviewer #2: Yes

3. Has the statistical analysis been performed appropriately and rigorously? 

Reviewer #2: Yes

4. Have the authors made all data underlying the findings in their manuscript fully available?

Reviewer #2: No

5. Is the manuscript presented in an intelligible fashion and written in standard English?

Reviewer #2: Yes

6. Review Comments to the Author

Reviewer #2: The authors rigorously adressed all my comments. Thank you

......................................................

7. PLOS authors have the option to publish the peer review history of their article (what does this mean?). If published, this will include your full peer review and any attached files.

Reviewer #2: **Yes: **Maxime Hentzien

---

## [Editor Report · Acceptance letter]

27 Aug 2024

PONE-D-23-11875R1 

PLOS ONE

Dear Dr. Champenois, 

I'm pleased to inform you that your manuscript has been deemed suitable for publication in PLOS ONE. Congratulations! Your manuscript is now being handed over to our production team.

Kind regards, 

on behalf of

Dr. Prakash Shakya 

Academic Editor

PLOS ONE